# Can Peri-Implant Marginal Bone Loss Progression and a-MMP-8 Be Considered Indicators of the Subsequent Onset of Peri-Implantitis? A 5-Year Study

**DOI:** 10.3390/diagnostics12112599

**Published:** 2022-10-26

**Authors:** Renzo Guarnieri, Rodolfo Reda, Alessio Zanza, Gabriele Miccoli, Dario Di Nardo, Luca Testarelli

**Affiliations:** 1Private Periodontal Implant Practice, 31100 Treviso, Italy; 2Department of Oral and Maxillofacial Sciences, Sapienza University of Rome, 00161 Rome, Italy; 3Department of Prosthodontics and Implantology, Saveetha Dental College and Hospitals, Saveetha Institute of Medical and Technical Sciences, Chennai 600077, India

**Keywords:** implant, MMP-8, marginal bone loss, peri-implant sulcus, interleukins

## Abstract

The aim of this retrospective study was to investigate the relationship between the amount of early bone remodeling, the marginal bone loss (MBL) progression, and the peri-implant sulcular fluid concentration of active metalloproteinase-8 (a-MMP-8) and the incidence of peri-implantitis (P) over 5 years of implant function. It has been documented that dental implants with a high degree of early marginal bone loss (MBL) are likely to achieve additional increased MBL during function. Moreover, it has been speculated that early increased MBL might be a predictive factor for the subsequent onset of peri-implant inflammatory diseases. Clinical and radiographic data at implant placement (T0) and restoration delivery (TR) at 6 months (T1), 2 years (T2), and 5 years (T5) post-loading were retrospectively collected. MBL levels/rates (MBLr) and peri-implant sulcular fluid levels/rates of a-MMP-8 were assessed at TR, T1, T2, and T5. Implants were divided into two groups: group 1 with peri-implantitis (P+) and group 2 without peri-implantitis (P−). A multi-level simple binary logistic regression, using generalized estimation equations (GEEs), was implemented to assess the association between each independent variable and P+. A receiver operating characteristics (ROC) curve was used to evaluate an optimal cutoff point for T1 MBL degree and a-MMP-8 level to discriminate between P+ and P− implants. A total of 80 patients who had received 80 implants between them (39 implants with a laser-microtextured collar surface (LMS) and 41 implants with a machined collar surface (MS)) were included. Periapical radiographs and a software package were used to measure MBL rates. Peri-implant sulcular implant fluid samples were analyzed by a chairside mouth-rinse test (ImplantSafe^®^) in combination with a digital reader (ORALyzer^®^). Twenty-four implants (six with an LMS and eighteen with an MS) were classified as P+. No statistically significant association was found between the amount of early bone remodeling, MBL progression, and MBLr and the incidence of peri-implantitis. Implants with a-MMP-8 levels >15.3 ng/mL at T1 presented a significantly higher probability of P+. The amount of early marginal bone remodeling cannot be considered as an indicator of the subsequent onset of P, whereas high a-MMP-8 levels 6 months after loading could have a distinct ability to predict P.

## 1. Introduction

In recent years, dental implantation has become a predictable procedure for partially or completely restoring edentulous patients with high success rates [1]. Nevertheless, with the increase in the use of dental implants, the incidence of peri-implant inflammatory diseases has also increased [2]. These diseases have been the topic of several consensus conferences, the most recent in 2017 [2], and they are classified as either peri-implant mucositis (PIM), in which inflammation is confined to the soft tissues [3], or peri-implantitis (P), in which the inflammatory process also extends to the supporting bone, with progressive loss beyond biological bone remodeling [4]. It is generally agreed that both PIM and P have an infectious etiology and that PIM usually precedes P [5]. However, the conversion from PIM to P remains an enigma, as P occurs primarily as a result of an overwhelming bacterial insult and subsequent host immune response [6]. The presence of pathogens is necessary but not sufficient for the development of P, since it is the osteo-immunoinflammatory mediators produced by the host response that exert an essential impact on the breakdown of peri-implant tissue. A recent literature review [7], summarizing the risk indicators for peri-implant diseases, determined that poor oral hygiene, a history of periodontitis and diabetes, and a lack of supportive peri-implant therapy are strongly associated with the development of peri-implant inflammatory diseases, while other potential risk indicators, such as smoking, a lack of keratinized tissue, and cement residue, have been identified with limited evidence.

It has been also hypothesized that a high degree of early marginal bone loss (MBL) might be a predictive factor for the subsequent onset of peri-implant inflammatory diseases [8,9,10,11]. Defining as “pathological” any MBL that exceeds the bone loss threshold during the healing/remodeling phase, several authors have highlighted that early increased MBL may be indicative of P development, as it may create a niche for pathogenic micro-organisms, providing a more anaerobic environment and promoting progressive bone loss [11,12].

The widely adopted thresholds for MBL are: 0.1–0.2 mm per year [13] or 2 mm [14] after the first year of loading. Other reported thresholds include: 2.5 mm bone loss after 5 years [15] and 1–1.5 mm [16] or 0.4 mm [17] from the time of loading. Although these bone-loss thresholds provide easy clinical ‘cutoffs’, they do not predict future MBL. Since marginal bone remodeling is a dynamic process, the rate of MBL has recently been proposed as a better index of implant success than bone-loss or bone-level values [8,9,10,11].

It is believed that the measurement of the concentration of a-MMP-8 in the peri-implant sulcular fluid (PISF) may be very helpful in detecting peri-implant tissues health or/and inflammatory status before clinical and radiographic measurements indicate pathologic changes [18,19,20,21,22]. Moreover, the determination of a-MMP-8 levels in PISF has been shown to be useful for screening susceptible sites and patients, differentiating peri-implant sites, and evaluating the progression of bone loss in peri-implantitis [23,24,25,26]. To date, the role played by the initial MBL degree in the development of P remains unknown [27]. Although some studies have reported that there is no association between the amount of initial physiological bone remodeling during the first year of implant placement and the incidence of P [10,11], a recent investigation of Lähteenmäki et al. [22] indicated that high a-MMP-8 levels in PISF were significantly more prevalent for dental implants with MBL > 2 mm.

Since peri-implant health can exist even in the presence of reduced bone support [28,29], it is important to determine whether the a-MMP-8 concentration in PISF, regardless of the degree of MBL, could represent a quantitative real-time chairside diagnostic test for the subsequent onset of P, and whether it could be used to assess the potential development and ongoing risk versus traditional clinical methods.

In a previous study [11], we retrospectively analyzed the MBL levels/rates and peri-implant sulcular fluid levels/rates of MMP-8 in 80 patients within three timeframes (6 months post-surgery (restoration delivery) and 6 and 24 months post-loading). The study had two main objectives: (1) to determine a possible cutoff point for discriminating between low- and high-bone-loss-type implants, considering a threshold of 2 mm at 24 months; and (2) to evaluate a possible correlation between peri-implant marginal bone loss progression and peri-implant sulcular fluid levels of a-MMP-8. The results showed that implants with increased MBL rates and a-MMP-8 levels at 6 months after loading were likely to exhibit additional marginal bone loss during the 2 years of follow-up. Moreover, the initial high levels of MBL and a-MMP-8 could be considered indicators of the subsequent progression of peri-implant MBL.

The present study was designed as a continuation of the previous study, to investigate in the same sample of patients whether a possible association between the incidence of peri-implantitis and the peri-implant MBL progression and PISF concentration of a-MMP-8 could be found after 5 years of implant function.

## 2. Materials and Methods

This study was approved by the Institutional Review Board of Sapienza University of Rome (prot. No. 4597) and conducted in accordance with the Helsinki Declaration. Inclusion criteria were: age > 18 years, the presence of at least one edentulous site in posterior areas, a physical status of I or II according to the American Society of Anesthesiologists (ASA) classification system, the absence of systemic diseases or conditions known to alter bone metabolism, and the presence of a stable periodontal condition. Exclusion criteria were: patients with incomplete charts, patients with a < 5-year follow-up period, patients with inaccessible files due to bad debt, patients with destroyed records, or patients who were deceased. As part of the data collection process, additional information was gathered at the time of implant placement, including age, tobacco usage and diabetic history, implant location, implant characteristics, collar surface, the mechanism of crown retention (screw- or cement-retained), the number of maintenance appointments, and the type of implant-abutment connection. For details regarding materials and methods, refer to the study by Guarnieri et al. [11], who previously reported comparative treatment outcomes of MBL progression and peri-implant sulcular fluid levels of a-MMP-8 at 6, 12, and 24 months post-loading.

Two groups of patients, treated with two different kinds of dental implants, were enrolled between January 2017 and January 2019. Group 1, comprising 41 patients, received 41 Tapered Internal TRX implants (BioHorizons, Birmingham, AL, USA). Group 2, comprising 39 patients, received 39 Tapered Internal TLX implants (BioHorizons, AL, USA). TRX and TLX implants have the same tapered macro design and the same grit-blasted body surface; TRX implants have a maximum coronal of 0.3 mm on the collar machined (M) surface, while TLX implants have a maximum coronal of 1.8 mm on the collar laser-microgrooved (LM) surface.

PISF sample analysis: the sampling site was prepared by removing excess saliva with a short, gentle blast of air. A sterile PISF collection strip was placed apically as deeply as possible into the sulcus at the sampling site using tweezers. The aMMP-8 levels were determined by the aMMP-8 PoC/chairside mouth rinse test (PerioSafe^®^) in combination with a digital reader (ORALyzer^®^), following the manufacturer’s instructions. This test is based on a lateral-flow sandwich immunoassay (DIPSTICK test) using the highly specific monoclonal antibodies MoAB 8706 and MoAB 8708, conjugated to latex particles.

Peri-implantitis definition: the presence of P was diagnosed according to the definition proposed by the American Academy of Periodontology/European Federation of Periodontology 2017 World Workshop on the Classification of Periodontal and Peri-implant Diseases and Conditions guidelines [5] and based on progressive bone loss beyond initial bone remodeling, increased probing depth compared to previous examinations, and the presence of bleeding and/or suppuration on gentle probing.

The radiographs were taken in high-resolution mode with a dental X-ray machine equipped with a long tube that operated at 70 Kw/7.5 mA, and specialized software (DBSWIN software, Durr Dental Italy S.r.l, Muggiò, Italy) was used for the linear measurement of marginal bone changes. A radiographic examination was performed at the time of implant placement (T0); at 6 months post-surgery (restoration delivery (TR)); and at 6 months (T1), 2 years (T2), and 5 years (T5) post-loading. If implants with peri-implantitis were treated, the last X-ray before treatment was considered as T2. At the same timepoints, the following clinical parameters were assessed at each implant site: the number of sites with plaque (P), the probing depth (PD), and the number of sites with bleeding on probing (BOP). In addition, the full-mouth plaque score (FMPS) and full-mouth bleeding score (FMBS) were also recorded. Peri-implant sulcus fluid samples were collected at T0, T1, T2, and T5 and analyzed for the MMP-8 concentration.

### Statistical Analysis

MBL and a-MMP-8 rates were calculated at T1, T2, and T5 by dividing the MBL and a-MMP-8 level by the number of months elapsed between the implant-placement and implant-loading stages. Three MBL rates (T1r, T2r, and T5r) were computed in millimeters/month (mm/m), and three a-MMP-8 rates (T1r, T2r, and T5r) were computed in nanograms/milliliters/months (ng/mL/m). The statistical analysis consisted of a description of categorical (absolute and relative frequencies) and continuous (mean, standard deviation, range, and median) variables for the total sample and differentiated between P+ and P− groups. At the implant level, a multi-level simple binary logistic regression using generalized estimation equations (GEE) was conducted to assess the association between each independent variable and P+ or P− diagnosis. Non-adjusted odds ratios (ORs) and 95% confidence intervals were obtained. Then, a combined model was estimated according to the relevant factors and covariates detected in the simple models. A linear regression model under the GEE approach was estimated to assess the correlation between MBL and a-MMP-8 rates and P+. The significance level used in the analysis was 5% (α = 0.05). A receiver operating characteristics (ROC) curve was used to explore an optimal cutoff point for T1 MBL and a-MMP-8 values to discriminate between P+ and P− implants. The area under the curve (AUC) and 95% confidence intervals were obtained. All statistical calculations were performed and figures created using Epitools Epidemiological Calculators (Ausvet; http://epitools.ausvet.com.au) (accessed on 9 June 2022).

## 3. Results

A total of 80 patients were included in the study (39 patients received 39 implants with a laser-microtextured collar surface, and 41 subjects received 41 implants with a machined/smooth surface). Table 1 reports the demographic/clinical parameters, implant follow-up, characteristics of patients and implants, prosthesis, and time protocols.

Table 2 reports the results for P, PD, BOP, FMPS, and FMBS recorded in the groups at the end of the follow-up period (T5).

At T5, no significant statistical difference was found in P, FMPS, and FMBS, whereas higher mean PD and BOP values were found in the P+ group (*p* < 0.05).

Overall, the included patient sample represented a total of 24 implants classed as P+ (6 with a laser-microtextured collar surface, and 18 with a machined/smooth surface). Table 3 reports the characteristics of the patients and implants, prosthesis, and time protocols by group.

Smoking habits, the implant collar surface type, the type of retention, and the years of follow-up showed statistically significant associations with the onset of P, while the other considered variables showed no associations. Table 4 reports the mean value of MBL and MBLr recorded in the groups.

At T1 and T2, no significant difference in MBL and MBLr were found between groups. At T5, the P+ group showed significantly higher MBL and MBLr values than the P− group.

Figure 1 and Figure 2 report the sensitivity and specificity and ROC analysis for MBL at T1 in the P+ group.

The results indicated that the level of MBL is not determined by chance. At T1, T2, and T6, a-MMP-8 levels and MMP-8 rates were higher in the P+ than the P− group, with a statistically significant difference. A summary of the data for the a-MMP-8 level at T1 is reported in Table 5.

Table 6 and Table 7 show the cutoff-point results for the target sensitivity and specificity of the a-MMP-8 level at T1, respectively.

Figure 3 shows the box plot of a-MMP-8 levels in the P+ group (infected) and P− group (uninfected).

Figure 4 and Figure 5 provide the sensitivity and specificity graph for the a-MMP-8 level at T1 and the area under the ROC curve for a-MMP-8 at T1 (95% CI for AUC), respectively.

The results indicated that a cutoff point of 15.3 ng/mL for a-MMP-8 has the distinct ability to predict the onset of P+ at 5 years.

## 4. Discussion

The results of our previous analysis [11], performed on the same sample at a follow-up of 2 years, showed that implants with increased MBL rates at 6 months after loading were likely to exhibit additional MBL. This correlation was confirmed by the present analysis, performed after 5 years of loading, which showed no statistically significant relationship between MBL progression and the onset of P. Similar results have been reported by Rodriguez et al. [10], who evaluated in a sample of 45 patients receiving 57 implants the possible relationship between the amount of early MBL and the presence of P after 1 year of loading. These data indicated that, although most of the implants presenting a high initial level of MBL exhibited progressively high MBL in the following years, the degree of MBL progression did not influence the onset of P. Some authors [8,9,10,11] have speculated that if MBL during the healing/remodeling phase exceeds a certain threshold, it may create a niche for pathogenic microorganisms, providing a more anaerobic environment and promoting P. Therefore, the MBL level could already be indicative and predictive of the development of P during the remodeling phase. The causes and risk factors associated with MBL have been extensively investigated [30], but an exploration of this topic was not the purpose of this paper. There is increasing evidence in the literature that MBL is generally an example of an imbalanced immunological reaction due to non-optimal implant components, surgery, prosthodontics, and/or compromised patient factors [31,32,33,34,35]. Bacteria are not needed to trigger bone resorption around a dental implant [36], but whether or not bacteria worsen the bone resorption is another issue. According to the bacterial theory, peri-implantitis is a chronic inflammatory condition associated with a microbial assault [2]. Although the etiology of P is bacterial, and some well-characterized pathogens present destructive virulence factors, the complex mechanisms of microbiota–environment–host interactions indicate that P is not a classical infection caused by one or a few “true” pathogens but is characterized by a polymicrobial breakdown of host homeostasis [37]. Recently, a polymicrobial synergy and dysbiosis model of disease development has been proposed [38]. The transition from host-compatible symbiotic to incipient dysbiotic microbiota involves an acute inflammatory immune response, which provides tissue-breakdown-derived nutrients for the bacteria, creating a self-perpetuating pathogenic cycle. This cyclic interaction in non-susceptible individuals may persist for years or, in susceptible patients, evolve rapidly into outright dysbiosis associated with an ineffective non-resolving inflammatory/immune response. Therefore, it is possible to hypothesize that patients presenting P could exhibit a compromised host response, leading to the presence of complex pathogenic dysbiotic microbiota that promote a dysregulated inflammatory response, resulting in the loss of peri-implant supporting tissues. Based in this concept, the peri-implant sulcus fluid has been tested for the presence of diverse molecules associated with the host-response–inflammatory pattern [39]. A-MMP-8, or collagenase-2, is considered one of the major mediators of peri-implant tissue destruction and the most prevalent collagenolytic protease in these diseased tissues [21,40]. The PICF level of a-MMP-8 was described as an early sign of peri-implant breakdown, related to the development of P around implants in response to plaque deposition [23,24,25,26]. In the present study, the PISF concentration of a-MMP-8 was assessed at the restoration delivery and 6 months, 2 years, and 5 years later. At each timepoint, the a-MMP-8 levels and a-MMP-8 rates in the P+ group were higher than in the P− group, with a statistically significant difference. Previous studies suggested a cutoff a-MMP-8 value of < 20 ng/mL to differentiate peri-implantitis sites from clinically healthy sites [18,19,20]. However, as far as the authors are aware, no data are available in the literature to support the ability of an a-MMP-8 threshold value to predict the onset of P. The receiver operating curve analysis indicated that the cutoff point of 15,3 ng/mL for a-MMP-8 at T1 (6 months post-loading) could have the ability to predict the onset of P. Almost all the implants with a-MMP-8 levels > 15.3 ng/mL at 6 months showed signs of P at 5 years. These data seem to suggest that the concentration of a-MMP-8 6 months following the restoration delivery has good reliability in identifying implants that could present P in the future.

The data collected in the current study demonstrate that the aMMP-8 chairside assay can be used as a convenient and reliable adjunctive tool in the early diagnosis of peri-implantitis before clinical and radiological signs document a full-blown lesion. The importance of the a-MMP-8 chairside diagnostic tool for clinicians lies in its ability to identify the destruction of collagenolytic tissue. Thus, it can be conveniently implemented to alert clinicians to and detect active collagenolysis affecting peri-implant tissues in the early stages of peri-implant disease. This could help clinicians more precisely tailor secondary prevention protocols based on the intensity of the rupture (collagenolytic activity) and, at the same time, improve patient compliance in terms of maintaining oral hygiene and adhering to appointments. There is a lack of longitudinal studies evaluating the correlation between the results of a-MMP-8 chairside tests and the evolution of P over time. In other words, no study has investigated the possible predictive diagnostic ability of a-MMP-8 in implant patients. Therefore, it is not possible to compare the results of the present study, and further investigations should be conducted to confirm our findings indicating the ability of a-MMP-8 chairside tests to predict the development of peri-implant disease over time in patients with no adverse clinical indices.

In the present analysis, the year of follow-up, retention type, smoking habits, and the implant collar surface type showed statistically significant associations with the onset of P. The negative influence of the year of follow-up on the onset of P has been previously well-documented in the literature [39,40]. Peri-implantitis progresses in a non-linear, accelerating pattern, and for most cases the onset occurs within 3 years of function [40]. A multilevel growth curve model revealed increased variance over time, which was attributed to subject heterogeneity [40]. Several observational studies [41] have reported on a correlation between excess cement and the prevalence of peri-implant diseases, indicating that the rough surface structure of cement remnants may facilitate retention and biofilm formation.

The deleterious effects of smoking on peri-implant tissue health have also been well-documented [41]. Smoking increases the expression and deposition of advanced glycation end-products in the peri-implant soft tissue, followed by the upregulation of pro-inflammatory cytokines, promoting tissue damage and alveolar bone resorption [42].

Regarding the collar surface, the results showed a higher incidence of P around implants with MS than around implants with LMS. These data agree with the results of previous experimental and clinical studies indicating that implants with LMS could have a lower predisposition to the onset of P compared to MS implants. It has been widely documented in the literature that an LMS on an implant collar/abutment allows the physical attachment of connective tissues, which presents high mechanical stability with the functional orientation of the connective fibers [43]. Therefore, it is possible to speculate that around LSM implants, the presence of an anatomical structure, likely of organized connective tissue, may play a protective role against the development of peri-implant inflammatory diseases.

It has been reported that smoking habits, general periodontal health, and gingival phenotype may be confounding factors, as they can influence aMMP-8 levels in PISF [44].

Another possible confounding factor is related to the standardization of PISF sampling due to the atypical morphology of the implant prosthesis. Inserting multiple strips of paper, cones, membranes, or other devices would be technically sensitive and could produce misleading results. Furthermore, it must be emphasized that due to the cyclic progression of peri-implant diseases, the biomarkers of the immune-inflammatory event responsible for tissue breakdown may not always be detected in cross-sectional studies with a single moment of fluid collection [38].

In addition to these limitations related to the aMMP-8 chairside test, the current investigation presents some other limitations. We did not consider the influence of soft tissue thickness, prosthetic abutment height, and emergence profiles on the development of P, which have been reported as important factors [44]. Moreover, the retrospective design and the limited sample of patients and implants mean that further studies are needed to confirm the results.

## 5. Conclusions

The amount of early marginal bone remodeling cannot be considered as an indicator of the subsequent onset of P, whereas high a-MMP-8 levels 6 months after loading could have a distinct ability to predict P.

## Figures and Tables

**Figure 1 diagnostics-12-02599-f001:**
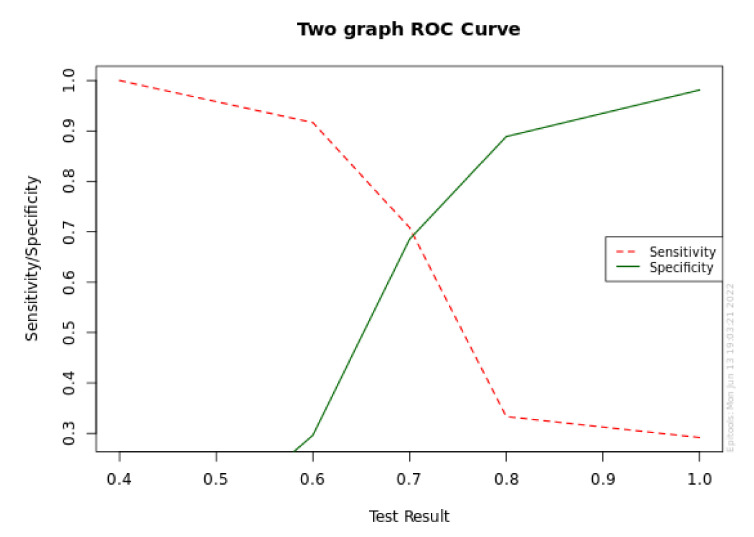
Sensitivity and specificity graph for MBL at T1.

**Figure 2 diagnostics-12-02599-f002:**
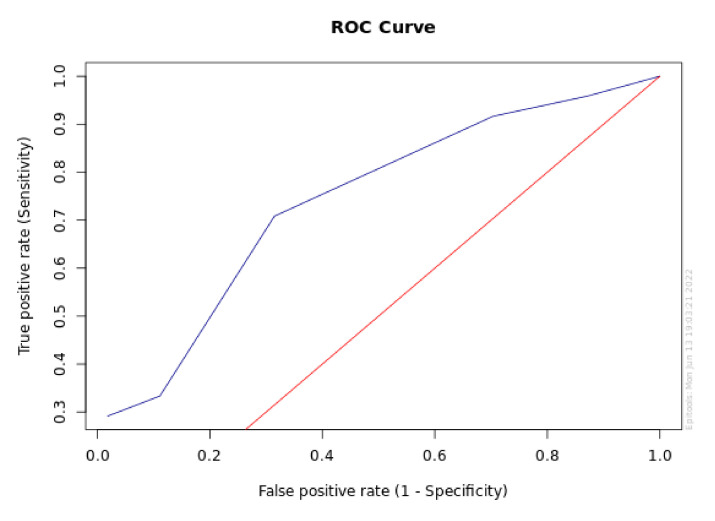
Area under the ROC curve for MBL at T1 (95% CI for AUC).

**Figure 3 diagnostics-12-02599-f003:**
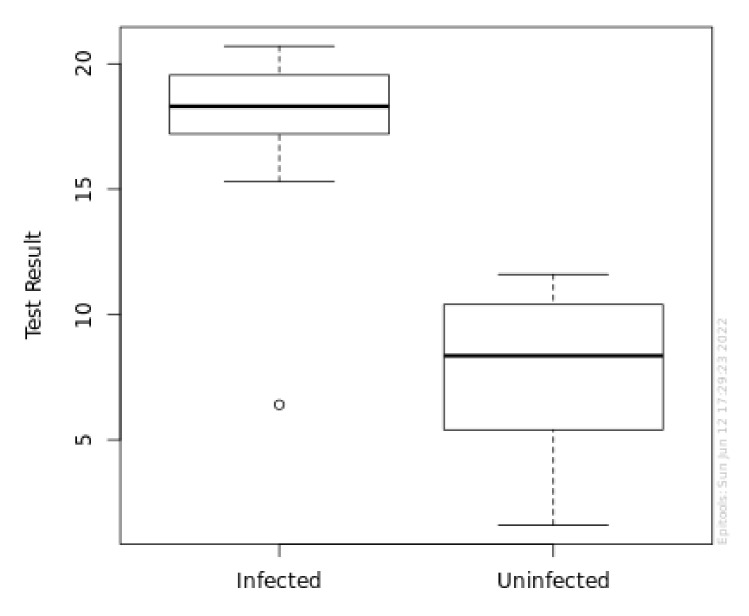
Box plot of a-MMP-8 levels in P+ group (infected) and P− group (uninfected).

**Figure 4 diagnostics-12-02599-f004:**
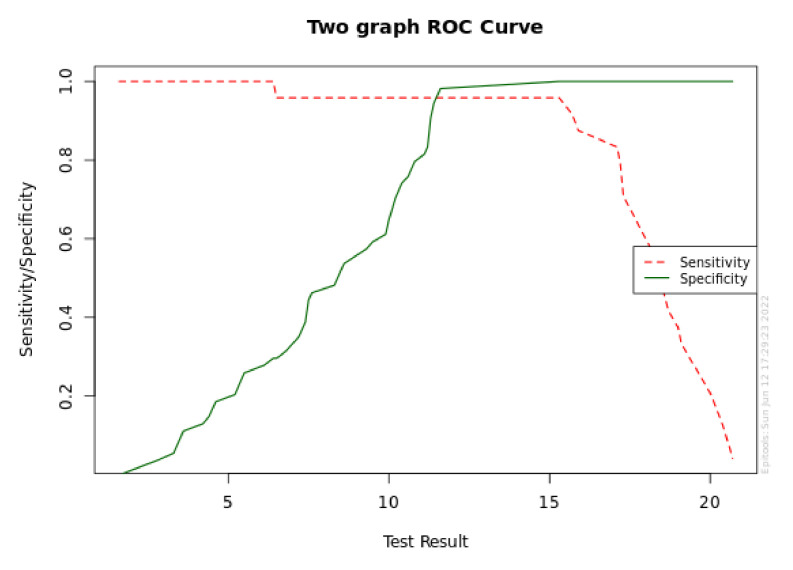
Sensitivity and specificity graph for a-MMP-8 level at T1.

**Figure 5 diagnostics-12-02599-f005:**
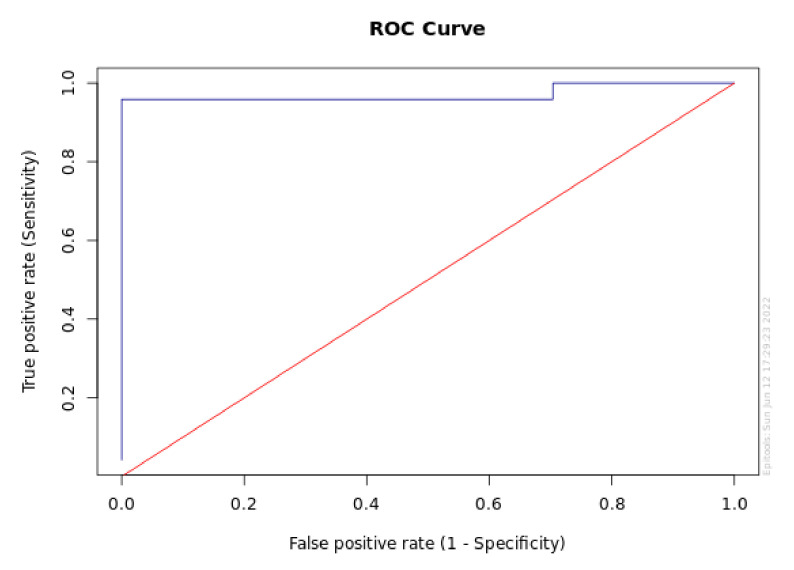
Area under the ROC curve for a-MMP-8 at T1 (95% CI for AUC).

**Table 1 diagnostics-12-02599-t001:** Demographic and clinical characteristics (%).

N. patients	80
N. implants	80
Age (years)	54.6 (9.6)
Gender:	
Male	38 (35)
Female	52 (65)
Smoking:	
No	60 (75)
Yes	20 (25)
Collar surface:	
LMS	41 (51.2)
MS	39 (48.8)
Location:	
Right mandible	16 (20)
Left mandible	22 (27.5)
Right maxilla 24 30	24 (30)
Left maxilla 18 22.5	18 (22.5)
Tooth Type:	
Molar	54 (67.5)
Premolar	26 (32.5)
Retention:	
Screwed	48 (60)
Cemented	32 (40)
Length (mm):	
<9	14 (17.5)
9–12	44 (55)
>12	22 (27.5)
Diameter (mm):	
<3.8	6 (7.5)
3.8–4.6	68 (85)
>4.6 mm	6 (7.5)
Follow-up (years):	
T1-T0	0.62 ± 0.2
T2-T1	1.92 ± 3.0
T5-T2	5.89 ± 3.8
T5-T0	9.64 ± 2.2.

**Table 2 diagnostics-12-02599-t002:** Mean values recorded at the end of the follow-up period (T5).

	N. Sites with Plaque	N. Sites with BOP	PD mm (SD)	FMPS (SD)	FMBS
P+	13	6	1.6 (08)	20.4 (2.1)	18.1 (1.3)
P−	18	24	5.8 (2.7)	19.8 (2.8)	19.5 (2.1)
Significance	>0.05	<0.05	<0.05	>0.05	>0.05

**Table 3 diagnostics-12-02599-t003:** Characteristics of patients and implants, prosthesis, and time protocols by P+/P− group: number of implants (%) or mean ± standard deviation. Results of simple binary logistic regression (odds ratio (OR) and 95% CI) using GEE model.

Parameter	P+ Group	P− Group	OR	95% CI	*p*-Value
N. implants	24 (30)	56 (70)			
Age	53.4 (7.2)	55.8 (9.4)	0.91	0.85–1.02	0.609
Gender:					
Male	10 (41.7)	26 (46.4)	1		
Female	14 (58.3)	30 (53.6)	1.04	0.31–1.06	0.546
Smoking:					
No	8 (33.3)	27 (48.2)	1		
Yes	16 (66.7)	29 (51.8)	1.80	0.24–2.68	0.072 *
Collar surface:					
LMS	6 (25)	35 (62.5)	1		
MS	18 (75)	21 (37.5)	1.18	1.02–1.19	0.031 *
Location:					
Right mandible	4 (16.6)	12 (21.4)	1		
Left mandible	6 (25)	16 (28.6)	1.06	0.33–7.95	0.769
Right maxilla	7 (29.2)	17 (30.3)	1.03	0.46–8.01	0.932
Left maxilla	7 (29.2)	11 (19.7)	1.11	0.34–9.01	0.789
Tooth Type:					
Molar	14 (58.3)	40 (71.4)	1		
Premolar	10 (41.7)	16 (28.6)	1.12	0.45–4.81	0.642
Retention:					
Screwed	6 (25)	32 (57.1)	1		
Cemented	18 (75)	14 (42.9)	1.72	0.81–3.44	0.037 *
Length (mm):					
<9	7 (29.1)	7 (12.5)	1		
9–12	12 (50)	32 (57.1)	1.10	0.36–2.36	0.451
>12	5 (20.1)	17 (30.4)	1.72	0.52–6.41	0.789
Diameter (mm):					
<3.8	1 (4.1)	5 (8.9)	1		
3.8–4.6	20 (62,5)	48 (85.7)	1.41	0.61–7.32	0.896
>4.6 mm	3 (33.4)	3 (5.4)	1.38	0.44–8.28	0.445
Follow-up (years):					
T1-T0	0.59 ± 0.8	0.65± 1.1	1		
T2-T1	1.90 ± 2.0	1.94 ± 1.2	1.22	0.81–1.56	0.464
T5-T2	4.92 ± 1.6	6.86 ± 1.7	1.64	1.45–2.45	0.034 *
T5-T0	8.72 ±2.4	10.56 ± 2.1	1.89	1.89–2.83	0.039 *

* Statistically significant.

**Table 4 diagnostics-12-02599-t004:** Mean value of MBL, MBL rates (MBLr), a-MMP-8, and a-MMP-8 rates (a-MMP-8r) recorded in the groups. MBL is expressed in mm, MBLr in mm/month, a-MMP-8 in nanograms/milliliters, and a-MMP-8r in nanograms/milliliters/month.

Parameter	P+ Group	P− Group	Significance
MBL at T1	0.89 ±0.53	0.75 ± 0.76	0.834
MBLr at T1	0.143 ± 0.12	0.125 ± 0.32	0.451
MBL at T2	1.52± 0.87	1.36 ± 0.65	0.212
MBLr at T2	0.063 ± 0.11	0.05± 0.06	0.322
MBL at T5	6.71 ± 1.8	2.23 ± 0.9	0.011 *
MBLr at T5	0.111 ± 0.31	0.039 ± 0.15	0.017 *
a-MMP-8 at T1	16.9 ± 9.4	7.6 ± 2.4	0.023 *
a-MMP-8r at T1	2.81 ± 1.9	1.2 ± 0.4	0.027 *
a-MMP-8 at T2	19.8 ± 7.3	11.4 ± 3.8	0.027 *
a-MMP-8r at T2	0.85 ± 0.9	0.21 ± 0.5	0.098 *
a-MMP-8 at T5	27.3 ± 9.1	12.3 ± 2.9	0.078 *
a-MMP-8r at T5	0.20 ± 0.8	0.02 ± 0.4	0.081 *

* Statistically significant.

**Table 5 diagnostics-12-02599-t005:** Summary of data for a-MMP-8 level at T1.

	Min	5%	25%	Median	75%	95%	Max	Mean	S. D.	Count
P+	6.4	15.4	17.2	18.3	19.5	20.4	20.7	17.8	2.87	24
P−	1.6	3.12	5.43	8.35	10.4	11.3	11.6	7.86	2.9	56

**Table 6 diagnostics-12-02599-t006:** Cutoff-point results for target sensitivity of a-MMP-8 level at T1.

Target Se	Cutoff point	Sensitivity	Se Lower 95% CL	Se Upper 95% CL	Specificity	Sp Lower 95% CL	Sp Upper 95% CL
0.999	6.4	1	0.862	1	0.296	0.191	0.428
0.995	6.4	1	0.862	1	0.296	0.191	0.428
0.99	6.4	1	0.862	1	0.296	0.191	0.428
0.98	6.4	1	0.862	1	0.296	0.191	0.428
0.95	15.3	0.958	0.798	0.993	1	0.934	1
0.9	15.7	0.917	0.742	0.977	1	0.934	1
0.8	17.1	0.833	0.641	0.933	1	0.934	1

**Table 7 diagnostics-12-02599-t007:** Cutoff-point results for target specificity of a-MMP-8 level at T1.

Target Sp	Cutoff point	Specificity	Sp Lower 95% CL	Sp Upper 95% CL	Sensitivity	Se Lower 95% CL	Se Upper 95% CL
0.999	15.3	1	0.934	1	0.958	0.798	0.993
0.995	15.3	1	0.934	1	0.958	0.798	0.993
0.99	15.3	1	0.934	1	0.958	0.798	0.993
0.98	11.6	0.981	0.902	0.997	0.958	0.798	0.993
0.95	11.6	0.981	0.902	0.997	0.958	0.798	0.993
0.9	11.3	0.907	0.801	0.96	0.958	0.798	0.993
0.8	11.1	0.815	0.692	0.896	0.958	0.798	0.993

## Data Availability

Not applicable.

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
