# Peer review of "Can Peri-Implant Marginal Bone Loss Progression and a-MMP-8 Be Considered Indicators of the Subsequent Onset of Peri-Implantitis? A 5-Year Study"

_diagnostics, 2022, doi:10.3390/diagnostics12112599_

Round 1
Reviewer 1 Report
the statistical package was not cited
Figures 3 and 4 shows the same information, box plot is suggested
ROC curves has limited usefulness for only 3 time analysis
the use of a chair side dignostic tool is interesting also limited to MMP-8, so you must give emphasis to this innovation and limitation....
Author Response
1) the statistical package was not cited
Response: in the revised manuscript the statistical package has been cited: “All statistical calculations were performed and figures were created using Epitools - Epidemiological Calculators (Ausvet. http://epitools.ausvet.com.au)
2) Figures 3 and 4 shows the same information, box plot is suggested
Response: in the revised manuscript Fig. 4 has been removed and Fig. 3 reported a box plot
3) ROC curves has limited usefulness for only 3 time analysis.
Response: authors agree with the consideration of the reviewer. However, Receiver Operating Characteristics (ROC) curve was used to explore an optimal cut-off point of T1 MBL and a-MMP-8 values to discriminate between P+ and P-implants. Moreover, in Discussion it has been added: …Furthermore, it must be emphasized that due to a cyclic progression of peri-implant diseases, the biomarkers of the immune-inflammatory event responsible for tissue breakdown may not always be active in cross-sectional studies with a single moment of fluid collection [44].
4) the use of a chair side diagnostic tool is interesting also limited to MMP-8, so you must give emphasis to this innovation and limitation....
Response: in the Discussion of the revised manuscript an emphasis to MMP-8 chair side diagnostic tool has been added:
“The data collected in the current study demonstrate that the aMMP‐8 chairside assay can be used as a convenient and reliable adjunctive tool in the early diagnosis of peri‐implantitis before the onset of clinical and radiological signs documents the full-blown lesion. The importance of the a-MMP-8 chair side diagnostic tool for clinicians is its ability to alarm the destruction of collagenolytic tissue. Thus, it can be conveniently implemented to alert for and detect active collagenolysis affecting peri‐implant tissues in the early stages of the peri-implant disease. This could help clinicians more precisely tailor secondary prevention protocols based on the intensity of the rupture (collagenolytic activity) and, at the same time, improve patient compliance in terms of maintaining oral hygiene and adherence to recall of patient’s appointments. There is a lack of longitudinal studies evaluating the correlation between the result of a-MMP-8 chair-side tests and the evolution of P over time. In other words, no study investigated a possible predictive diagnostic ability of a-MMP-8 in implant patients. Therefore, it is not possible to compare the results of the present study, and further investigations should be conducted to confirm our findings indicating the ability of a-MMP-8 chair-side tests to predict the development of peri-implant disease over time in patients with no adverse clinical indices…
… It has been reported that smoking habits, general periodontal health and gingival phenotype may be confounding factors, as they can influence aMMP-8 levels in PISF [44].
Another possible confounding factor is related to the standardization of PISF sampling due to the atypical morphology of the implant prosthesis. Inserting multiple strips of paper, cones, membranes, or other devices would be technically sensitive and could produce misleading results. Furthermore, it must be emphasized that due to a cyclic progression of peri-implant diseases, the biomarkers of the immune-inflammatory event responsible for tissue breakdown may not always be active in cross-sectional studies with a single moment of fluid collection [44].
In addition to these limitations related to the aMMP‐8 chairside test, the current investigation presents some other limitations. It has not considered the influence of soft tissue thickness, prosthetic abutment height and emergence profiles on development of P, which have been reported as important factors [45]. Moreover, the retrospective design and the limited sample of patients and implants, request further study to confirm the results.
Reviewer 2 Report
Abstract. This section is correct and show the summary of the paper.
Introduction. This section is correct, but the authors, in the last paragraph, must report clinical finding of others studies. In fact, only shows the results of a previous similar study of the same group.
Moreover, the authors must be report the influence of cervical surface implant in MBL and peri-implantitis, because the study assesses two different surfaces.
Material and methods.
The structure of this section must be improved. The authors must report the type of implants and the used methodology for a-MMP-8.
Results. The structure of this section including tables and figures, is correct.
Smoking habit, the kind of collar surface of implant, the type of retention and year of follow-up showed a statistically significant association, while others considered variables showed no association. Results indicated that the distinct ability of the MBL is not different from chance. At T1, T2 and T6, a-MMP-8 levels and MMP-8 rates were higher in P+ than in P- group, with a statistically significant difference.
Discussion. This section must report the analysis of the results and it comparison with other studies.
The authors must report more discussion in the paragraph between 272-280 lines (i.e. smoking habits).
Conclusions. This section is not correct. The limitations of the paper must be changed to the end of discussion. In this section the authors report the main aspects of the paper
Conclusively, the study is not ready for publication.
Author Response
Abstract. This section is correct and show the summary of the paper.
Introduction. This section is correct, but the authors, in the last paragraph, must report clinical finding of others studies. In fact, only shows the results of a previous similar study of the same group.
Response: in the revised manuscript clinical finding of others study have been reported:
“To date, the role played from the initial MBL degree in the development of P remains unknown. Although some studies have reported that there is not association between the amount of initial physiological bone remodeling during the first year of implant placement and incidence of P [10,11], a recent investigation of Lähteenmäki et al. [22] indicated that high a-MMP-8 levels in PISF were significantly more prevalent for dental implants with MBL> 2 mm.
Since peri-implant health can exist even in the presence of reduced bone support [28,29], it is important to determine whether the a-MMP-8 concentration in PISF, regardless of the degree of MBL present, could represent a quantitative real-time chairside diagnostic test for the subsequent onset of P, as well as to assess the potentially developing and ongoing risk versus traditional clinical methods.”
Moreover, the authors must be report the influence of cervical surface implant in MBL and peri-implantitis, because the study assesses two different surfaces.
Response: in the revised manuscript, the influence of cervical surface implant in MBL and peri-implantitis has been discussed in “Discussion”
Material and methods.
The structure of this section must be improved. The authors must report the type of implants and the used methodology for a-MMP-8.
Response: in the revised manuscript in Materials and Methods it has been added: “Two groups of patients, treated with two different kinds of dental implants were enrolled between January 2017 and January 2019. Group 1, comprising 41 patients, received 41 Tapered Internal TRX implants (BioHorizons, Birmingham, AL, USA). Group 2, comprising 39 patients, received 39 Tapered Internal TLX (BioHorizons, AL, USA). TRX and TLX implants have the same tapered macro design and the same body grit-blasted surface; TRX implants have a maximum coronal of 0.3 mm on the collar machined (M) surface, while TLX implants have a maximum coronal of 1.8 mm on the collar laser-microgrooved (LM) surface.
PISF sample analysis: the sampling site was prepared by means of the removal of excess saliva with a short, gentle blast of air. A sterile PISF collection strip was placed apically as deeply as possible into the sulcus at the sampling site using tweezers. aMMP-8 levels were determined by the aMMP-8 PoC/chairside mouth rinse test, (PerioSafe®), in combination with a digital reader, (ORALyzer®), following the manuacturer’s instructions [32,38]. This test is based on a lateral flow sandwich immunoassay (DIPSTICK test) using the highly specific monoclonal antibodies MoAB 8706 and MoAB 8708, conjugated to latex particles.
Results. The structure of this section including tables and figures, is correct.
Smoking habit, the kind of collar surface of implant, the type of retention and year of follow-up showed a statistically significant association, while others considered variables showed no association. Results indicated that the distinct ability of the MBL is not different from chance. At T1, T2 and T6, a-MMP-8 levels and MMP-8 rates were higher in P+ than in P- group, with a statistically significant difference.
Discussion. This section must report the analysis of the results and it comparison with other studies. The authors must report more discussion in the paragraph between 272-280 lines (i.e. smoking habits).
Response: in the revised manuscript, in Discussion, it has been added:
“The data collected in the current study demonstrate that the aMMP‐8 chairside assay can be used as a convenient and reliable adjunctive tool in the early diagnosis of peri‐implantitis before the onset of clinical and radiological signs documents the full-blown lesion. The importance of the a-MMP-8 chair side diagnostic tool for clinicians is its ability to alarm the destruction of collagenolytic tissue. Thus, it can be conveniently implemented to alert for and detect active collagenolysis affecting peri‐implant tissues in the early stages of the peri-implant disease. This could help clinicians more precisely tailor secondary prevention protocols based on the intensity of the rupture (collagenolytic activity) and, at the same time, improve patient compliance in terms of maintaining oral hygiene and adherence to recall of patient’s appointments. There is a lack of longitudinal studies evaluating the correlation between the result of a-MMP-8 chair-side tests and the evolution of P over time. In other words, no study investigated a possible predictive diagnostic ability of a-MMP-8 in implant patients. Therefore, it is not possible to compare the results of the present study, and further investigations should be conducted to confirm our findings indicating the ability of a-MMP-8 chair-side tests to predict the development of peri-implant disease over time in patients with no adverse clinical indices…
…Peri-implantitis progresses in a non-linear, accelerating pattern and for most cases the onset occurs within 3 years of function [40]. A multilevel growth curve model revealed an increased variance over time that was attributed to subject heterogeneity [40]. Several observational studies [41] have reported on a correlation between excess cement and the prevalence of peri-implant diseases, indicating that the rough surface structure of cement remnants may facilitate retention and biofilm formation.
The deleterious effects of smoking on peri-implant tissue health have also been well documented [41]. Smoking increases the expression and deposition of advanced glycation end products in the peri-implant soft tissue followed by the upregulation of pro-inflammatory cytokines, promoting tissue damage and alveolar bone resorption [42]…
…It has been reported that smoking habits, general periodontal health and gingival phenotype may be confounding factors, as they can influence aMMP-8 levels in PISF [44].
Another possible confounding factor is related to the standardization of PISF sampling due to the atypical morphology of the implant prosthesis. Inserting multiple strips of paper, cones, membranes, or other devices would be technically sensitive and could produce misleading results. Furthermore, it must be emphasized that due to a cyclic progression of peri-implant diseases, the biomarkers of the immune-inflammatory event responsible for tissue breakdown may not always be active in cross-sectional studies with a single moment of fluid collection [44]. In addition to these limitations related to the aMMP‐8 chairside test, the current investigation presents some other limitations. It has not considered the influence of soft tissue thickness, prosthetic abutment height and emergence profiles on development of P, which have been reported as important factors [45]. Moreover, the retrospective design and the limited sample of patients and implants, request further study to confirm the results.
Conclusions. This section is not correct. The limitations of the paper must be changed to the end of discussion. In this section the authors report the main aspects of the paper
Response: In the revised manuscript, it has been reported in Conclusions:
“The amount of early marginal bone remodeling cannot be considered as indicators of the subsequent onset of P, whereas high a-MMP-8 levels, 6 months after loading, could have the distinct ability to predict P.
Conclusively, the study is not ready for publication.